

# Simple proxies for estimating the concentrations of monoterpenes and their oxidation products at a boreal forest site

Jenni Kontkanen[1], Pauli Paasonen[1], Juho Aalto[2,3], Jaana Bäck[3], Pekka Rantala[1], Tuukka Petäjä[1], and Markku Kulmala[1]

[1]Department of Physics, University of Helsinki, P.O. Box 64, FIN-00014 University of Helsinki, Finland
[2]Hyytiälä Forestry Field Station, Hyytiäläntie 124, FIN-35500 Korkeakoski, Finland
[3]Department of Forest Sciences, University of Helsinki, P.O. Box 27, FIN-00014 University of Helsinki, Finland

*Correspondence to*: Jenni Kontkanen (jenni.kontkanen@helsinki.fi)

**Abstract.** The oxidation products of monoterpenes likely have a crucial role in the formation and growth of aerosol particles in boreal forests. However, the continuous measurements of monoterpene concentrations are usually not available in decadal time scales, and the direct measurements of the concentrations of monoterpene oxidation product are so far scarce. In this study we developed proxies for the concentrations of monoterpenes and their oxidation products at a boreal forest site in Hyytiälä, southern Finland. For deriving the proxies we used the monoterpene concentration measured with a proton transfer reaction mass spectrometer (PTR-MS) during 2006–2013. Our proxies for the monoterpene concentration take into account the temperature-controlled emissions from the forest ecosystem, the dilution caused by the mixing within the boundary layer, and different oxidation processes. All the versions of our proxies captured the seasonal variation of the monoterpene concentration, the typical proxy-to-measurements ratios being between 0.8 and 1.3 in summer and between 0.6 and 2.6 in winter. In addition, the proxies were able to describe the diurnal variation of the monoterpene concentration rather well, especially in summer months. By utilizing one of the proxies, we calculated the concentration of oxidation products of monoterpenes by considering their production in the oxidation and their loss due to condensation on aerosol particles. The concentration of oxidation products was found to have a clear seasonal cycle with the maximum in summer and the minimum in winter. The concentration of oxidation products was lowest in the morning or around noon and highest in the evening. In the future, our proxies for the monoterpene concentration and their oxidation products can be used, for example, in the analysis of new particle formation and growth in boreal environment.

## 1 Introduction

Terrestrial ecosystems emit large amounts of biogenic volatile organic compounds (BVOCs) into the atmosphere (Guenther et al., 2012), where they are oxidized forming less volatile vapors. In boreal forests BVOC emissions are typically dominated by monoterpenes (Hakola et al., 2006; Rinne et al., 2009). Recent studies have shown that the low volatile oxidation products of monoterpenes may participate in atmospheric particle formation and growth, and thus affect the aerosol-radiation interactions and the concentrations of cloud condensation nuclei in the atmosphere (Kulmala et al., 1998; O'Dowd et al., 2002; Kulmala





et al., 2013; Paasonen et al., 2013; Ehn et al., 2014; Jokinen et al., 2015). Therefore, the knowledge of the concentrations of monoterpenes and their oxidation products is crucial when estimating the climate effects of aerosol particles.

The total concentration of monoterpenes in the boundary layer can be measured using online techniques such as a proton transfer reaction mass spectrometer (PTR-MS; Taipale et al., 2008), or by collecting air samples and analyzing them with gas
chromatography which also separates the different monoterpenes from each other (Hakola et al., 2003). Monoterpene concentration in the boreal forest has been observed to be lowest in winter (below 0.1 ppbv) and highest in summer (>0.25 ppbv) (Hakola et al., 2009; Lappalainen et al., 2009). The summertime maximum in the concentration results from the fact that the emissions of monoterpenes from the vegetation are highest in summer, as they are mainly controlled by temperature and linked to plant activity (Tarvainen et al., 2005; Hakola et al., 2006; Rantala et al., 2015). In some studies measured
monoterpene concentrations have been used for estimating the concentration of their oxidation products (Lehtipalo et al., 2011). However, monoterpene concentration data is often available only for short measurement periods, and thus they are not always suitable for the analysis of long-term data sets. Furthermore, the recent development of chemical ionization mass spectrometry techniques has enabled the direct measurements of monoterpene oxidation products but these data are still very scarce (Ehn et al., 2014; Jokinen et al., 2015).

Due to the limited amount of data on the concentrations of monoterpenes and their oxidation products, in some studies they have been estimated by using simple proxies. One proxy for monoterpene concentration was obtained by parametrizing measured monoterpene concentration as a function of air temperature (Lappalainen et al., 2009). This proxy has been utilized for calculating the oxidation products of monoterpenes from reactions with hydroxyl radical (OH) and ozone ($O_3$) (Nieminen et al., 2014). However, this earlier approach has several limitations: 1) only daytime values of measured monoterpene
concentration were used for the parametrization; 2) the mixing within the boundary layer, diluting monoterpene concentration, was not considered; 3) the oxidation of monoterpenes by nitrate radical ($NO_3$), a major loss mechanism of monoterpenes at night (Peräkylä et al., 2014; Mogensen et al., 2015), was not included. Therefore, this proxy is not able to describe the diurnal variation of the concentrations of monoterpenes and their oxidation products.

In this study, we construct improved proxies for the concentration of monoterpenes and their oxidation products at a boreal
forest site in Hyytiälä, southern Finland. Our proxies for monoterpene concentration include both biological, physical, meteorological, and chemical processes: the temperature-driven emissions of monoterpenes, the dilution of the concentration caused by the mixing within the boundary layer, and the oxidation of monoterpenes by $O_3$, OH and $NO_3$. For deriving these proxies we use monoterpene concentration measured in Hyytiälä during 2006–2013. To assess the performance of the novel proxies, we compare different versions of the proxy to the measured monoterpene concentration, and investigate how well the
proxies are able to describe the observed seasonal and diurnal variation of monoterpene concentration. Finally, we use one of the monoterpene proxies to calculate the concentration of the oxidation products of monoterpenes in Hyytiälä during 1996–2014 and investigate its seasonal and diurnal cycle.



## 2 Material and methods

### 2.1 Measurements

The measurements were performed during 2006–2013 at the SMEAR II station in Hyytiälä, southern Finland (Hari and Kulmala, 2005). The station is located in the southern boreal vegetation zone, and it is surrounded by a rather homogeneous

Scots pine (*Pinus sylvestris*) forest (Ilvesniemi et al., 2009; Bäck et al., 2012).

For constructing the proxy for monoterpene concentration, we used the volume mixing ratios of monoterpenes measured with a proton transfer reaction quadrupole mass spectrometer (PTR-MS; Ionicon Analytik BmbH, Austria) (Taipale et al., 2008). The measurements were conducted close to the forest canopy at the height of 14 m (2006–2009) or 16.8 m (2010–2013) (Taipale et al., 2008; Rantala et al., 2015). Until March 2007 the measurements were performed every second hour and after

that every third hour. The measurements were not conducted continuously during the years 2006–2013 but, especially in the beginning, only during intensive measurement campaigns. The number of data points (1-hour averages) obtained for each month are presented in Table 1, which shows that there are more data available in summer and spring months than in autumn and winter. To reduce the effect of anthropogenic pollution episodes on monoterpene concentration, the data during time periods when the wind direction corresponded to the direction of the nearby sawmill were omitted from the analysis (Liao et

al., 2011).

For calculating the proxy, the concentrations of ozone ($O_3$) and nitrogen oxides (NO and $NO_x$) were utilized. $O_3$ concentration was recorded with an ozone analyzer (TEI 49C, Thermo Fisher Scientific, Waltham, MA, USA) based on the absorption of UV radiation. NO and $NO_x$ concentrations were measured with a chemiluminescence analyzer (TEI 42C TL, Thermo Fisher Scientific, Waltham, MA, USA). $NO_2$ concentration was calculated by subtracting NO concentration from $NO_x$ concentration.

The 30-minute averages of the $O_3$, NO, and $NO_2$ concentration measured at the height of 16.8 m were used in the analysis.

In addition, the 30-minute averages of UV-B radiation, temperature and wind speed were used in the calculations. UV-B radiation was measured with a SL 501A pyranometer (Solar Light, Philadelphia, PA, USA) at the height of 18 m. Temperature was measured with a PT-100 sensor at 16.8 m height. Wind speed was measured at the height of 16.8 m with a cup anemometer (A101M/L, Vector Instruments, Rhyl, Clwyd, UK) until September 2003 and with an ultrasonic anemometer (Ultrasonic

anemometer 2D, Adolf Thies GmbH, Göttingen, Germany) after that. Furthermore, ECMWF (European Centre for Medium-Range Weather Forecasts) reanalysis data were used for determining the boundary layer height (BLH) in Hyytiälä.

Finally, when calculating the oxidation products of monoterpenes, condensation sink (CS), describing the loss rate of vapors due to the condensation on aerosols particles, was calculated from the particle size distribution data (Kulmala et al., 2001). Particle size distributions were measured with a Differential Mobility Particle Sizer (DMPS; Aalto et al., 2001) at the ground

level.



### 2.2 Proxy calculations

#### 2.2.1 Proxy for monoterpenes

The concentration of monoterpenes in the boundary layer is determined by various physical, chemical, meteorological and biological processes. In Hyytiälä the main source of monoterpenes is the emissions from the forest ecosystem, which are largely controlled by air temperature (Guenther et al., 1993). The most important sink of monoterpenes is their oxidation by $O_3$, OH and $NO_3$ radicals (Atkinson and Arey, 2003). In addition, the monoterpene concentration is strongly affected by dilution caused by the mixing of the boundary layer.

For monoterpene emissions we used a temperature ($T$) -dependent exponent function, which has been observed to describe the monoterpene emissions well in Hyytiälä (Tarvainen et al. 2005; Lappalainen 2009):

$$E = \alpha \times \exp(\beta(T - T_s)) \tag{1}$$

Here $\alpha$ and $\beta$ are empirical parameters and $T_s$ is 303.15 K.

The sink of monoterpenes due to oxidation by $O_3$, OH and $NO_3$ can be calculated from

$$S_{oxidation} = k_{OH+MT}[OH] + k_{O3+MT}[O_3] + k_{NO3+MT}[NO_3] \tag{2}$$

Here $k_{OH+MT}$, $k_{O3+MT}$ and $k_{NO3+MT}$ are reaction rate coefficients between monoterpenes and different oxidants. To obtain the correct diurnal cycle for the reaction rate coefficients, we used temperature-dependent relations for alpha-pinene (Atkinson et al., 2006; see Table A1 in Appendix). Alpha-pinene is the most abundant monoterpene in Hyytiälä during summer but also delta-3-carene, camphene, limonene and beta-pinene contribute significantly to the total monoterpene concentration (Hakola et al., 2009; Bäck et al., 2012; Hakola et al., 2012). In winter, camphene has, on average, the highest concentration, followed by alpha-pinene (Hakola et al., 2012). To take into account the seasonal variation of reaction rate coefficients caused by the changes in the composition of monoterpenes, we utilized the monthly-mean reaction rate coefficients presented by Peräkylä et al. (2014).

$O_3$ is the only oxidant in Eq. (2) having its concentration directly measured in Hyytiälä. The concentration of OH was calculated by scaling the measured UV-B-radiation with the empirically derived factors from Petäjä et al. (2009):

$$[OH]_{proxy} = \left(\frac{8.4 \times 10^{-7}}{8.6 \times 10^{-10}} UVB^{0.32}\right)^{1.92} \tag{3}$$

Measuring of $NO_3$ concentration is challenging and has been conducted in Hyytiälä only for a short time period during which $NO_3$ mixing ratios were mostly below the detection limit of the instrument (Williams et al., 2011; Mogensen et al., 2015). Therefore, we estimated the concentration of $NO_3$ in a similar way as was done by Peräkylä et al. (2014). A steady state between the production of $NO_3$ in the reaction between $O_3$ and $NO_2$ and the removal of $NO_3$ was assumed:

$$[NO_3] = k_{O3+NO2}[O_3][NO_2] \times \tau_{NO3}. \tag{4}$$





Here $k_{O3+NO2}$ is the temperature-dependent reaction rate coefficient between $NO_2$ and $O_3$, which was calculated from a temperature-dependent relation (Atkinson et al., 2004; see Table A1). $\tau_{NO3}$ is the lifetime of $NO_3$.

During daytime $NO_3$ is efficiently removed in the photolysis, and thus we assumed for it a lifetime of 5 s for all times when UV-B radiation was higher than 0.01 W m$^{-2}$ (Peräkylä et al., 2014). The lifetime during nighttime was calculated from

$$5 \quad (\tau_{NO3})^{-1} = k_{NO3+MT}[MT] + k_{NO3+NO}[NO] + (k_{N2O5+H2O}[H_2O]) \times K \, [NO_2]. \tag{5}$$

Here $k_{NO3+MT}$ is the reaction rate coefficient between monoterpenes and $NO_3$, and $k_{NO3+NO}$ between $NO_3$ and NO, for which temperature-dependent relations were used (Atkinson et al., 2004, 2006; see Table A1). $K$ is the equilibrium constant for the reaction between $NO_3$ and $NO_2$ producing $N_2O_5$, which was calculated from the relation $K = 5.1 \times 10^{-27} \exp(10871/T)$ (Osthoff et al., 2007). $k_{N2O5+H2O}$ is the reaction rate coefficient between $N_2O_5$ and water vapor for which the value of $2.5 \times 10^{-22}$ cm$^3$ s$^{-1}$ was used (Atkinson et al., 2004). In reality, $NO_3$ reacts also with other VOCs, such as isoprene, but in a pine forest in Hyytiälä their contribution to the lifetime is only minor compared to the reactions with monoterpenes (Peräkylä et al., 2014). Furthermore, when calculating the lifetime of $NO_3$, Peräkylä et al. (2014) also considered the heterogeneous uptake of $N_2O_5$ by aerosol particle surfaces but we omitted that process from our calculations due to its minor effect on the lifetime according to their study.

By combining Eqs (1) and (2), and including the effect of the mixing within the boundary layer by using the mixing layer height (BLH) and wind speed (ws), the equation for the ideal monoterpene proxy, including all the processes, can be written as:

$$[MT]_{proxy,ideal} = \frac{a \exp(b(T-T_s))}{k_{OH+MT}[OH] + k_{O3+MT}[O_3] + k_{NO3+MT}[NO_3]} \times f(BLH) \times f(ws). \tag{6}$$

The values for empirical parameters a and b and the functional forms of f(BLH) and f(ws) were determined as follows. First, an initial value for the parameter b, 0.09 K$^{-1}$, was obtained from the literature (Guenther et al., 1993). The BLH dependence was then inspected by plotting the ratio of the measured monoterpene concentration to the calculated steady-state concentration (the first term on the right hand side of Eq. (6)) as a function of BLH. The different forms of dependences between the ratio and BLH were tested, and the power-law form f(BLH)=BLH$^c$ (for BLH values above 100 m) was found to describe the relation best (Fig. 1a). Next, the ratio of the measured monoterpene concentration to the product of the term BLH$^c$ (the value for c was fitted as in Fig 1a, for BLH<100 m the value of 100$^c$ was used) and the steady state concentration (the first term on the right hand side of Eq. (6)) was depicted against wind speed (Fig 1b). The power-law form was found to be most suitable also for this dependence and an initial value for the parameter d in f(ws)=ws$^d$ was extracted from the fitting. The effect of RH was tested in a similar manner, but no dependence was found. Consequently, the equation for the ideal form of the monoterpene proxy becomes:

$$30 \quad [MT]_{proxy,ideal} = \frac{a \exp(b(T-T_s))}{k_{OH+MT}[OH] + k_{O3+MT}[O_3] + k_{NO3+MT}[NO_3]} \times BLH^c \times ws^d. \tag{7}$$



After determining the initial values for the parameters b, c and d, they were optimized by minimizing the variability of the data-point specific ratios between the proxy and the measurements, with the method presented by Paasonen et al. (2010). The variability was determined as the ratio between 90th and 10th percentiles ($V_{90/10}$) of the proxy to measurement ratios (see an illustration of the meaning of $V_{90/10}$ in Fig. 2). The optimization was done with the Matlab-function fminsearch by searching for the values of b, c and d yielding the smallest variability, i.e. $V_{90/10}$. The initial values for b, c and d in the fminsearch-script were set to the values determined as described after Eq. (6). However, we also varied the initial values to see if the obtained parameters would present a local minimum in the variability. Finally, the value for the parameter a was determined as the geometric mean value of the ratio between the measured and proxy concentrations. The values obtained for the empirical parameters are presented in Table 2. Note that we used only data recorded during March–November for determining the parameters, thus excluding the data from winter months when biogenic emissions of monoterpenes are low. For finding the optimal parameters, we chose to minimize $V_{90/10}$ instead of, for example, maximizing the correlation coefficient, because with the chosen method the proxy concentrations are optimized towards one-to-one response to the measured concentrations. A higher value for the correlation coefficient between the proxy and measurements could be obtained with some other parameter values, but they might lead to a physically unsound non-linear dependence between the proxy and measurements.

The disadvantage of the ideal version of the proxy presented by Eq. (7) is that monoterpene concentration is needed for calculating $NO_3$ concentration. Thus, this proxy is not useful in reality, as it cannot be used for estimating monoterpene concentration for times without monoterpene measurements. In order to overcome this problem, we modified the ideal proxy in terms of how $NO_3$ concentration is dealt with. The simplest way is to neglect the oxidation of monoterpenes by $NO_3$, in which case the proxy becomes

$$[MT]_{proxy1} = \frac{a_1 \exp(b_1(T-T_s))}{k_{OH+MT}[OH] + k_{O3+MT}[O_3]} \times BLH^{c_1} \times ws^{d_1} . \qquad (8)$$

Here $a_1$, $b_1$, $c_1$, and $d_1$ are empirical parameters, which were determined as explained above (see Table 2).

The oxidation of monoterpenes by $NO_3$ presents a significant loss for monoterpenes during nighttime in Hyytiälä (Peräkylä et al., 2014; Mogensen et al., 2015) and therefore this mechanism should ideally be included in the monoterpene proxy. Thus, we tested a proxy in which we used a constant value of $4.3 \times 10^9$ cm$^{-3}$ for monoterpene concentration when calculating the lifetime of $NO_3$ from Eq. (5). The constant value was obtained by calculating the median of monoterpene concentration measured at night. In this way, the second version of the proxy was obtained, now including the oxidation by $NO_3$:

$$[MT]_{proxy2} = \frac{a_2 \exp(b_2(T-T_s))}{k_{OH+MT}[OH] + k_{O3+MT}[O_3] + k_{NO3+MT}[NO_3(MT_{const})]} \times BLH^{c_2} \times ws^{d_2} \qquad (9)$$

The values of the empirical parameters $a_2$, $b_2$, $c_2$, and $d_2$ are presented in Table 2.

Finally, we applied an iterative method and calculated the $NO_3$ lifetime by using the monoterpene proxy obtained from Eq. (9). This way, we obtained the third version of the monoterpene proxy:

$$[MT]_{proxy3} = \frac{a_3 \exp(b_3(T-T_s)}{k_{OH+MT}[OH] + k_{O3+MT}[O_3] + k_{NO3+MT}[NO_3(MT_{iter})]} \times BLH^{c_3} \times ws^{d_3} \qquad (10)$$





The values of the empirical parameters $a_3$, $b_3$, $c_3$, and $d_3$ are shown in Table 2.

Additionally, we tested a simplified version of the proxy by including only the monoterpene emissions and the mixing of the boundary layer, and omitting the sink due to oxidation. In this case the proxy becomes:

$$[MT]_{proxy,simple} = a_4 \exp(b_4(T - T_s)) \times BLH^{c_4} \times ws^{d_4} . \tag{11}$$

The values of the empirical parameters $a_4$, $b_4$, $c_4$, and $d_4$ are presented in Table 2.

It needs to be noted that because of the interrelations between the diurnal and annual cycles of temperature, BLH, wind speed and the concentrations of OH, $O_3$ and $NO_3$, the values obtained for the empirical parameters depend always on the other factors in the proxy. For example, the value optimized for the parameter $d_4$ in the simplest version of the proxy differs significantly from the parameters d derived for the other proxies (see Table 2) presumably because of the unaccounted diurnal and/or
10 seasonal cycles of the oxidant concentration in the simple proxy.

Furthermore, when studying the correlation between the proxies and measurements (see Sect. 3.1.1), we observed that $MT_{proxy1}$, obtained from Eq. (8), often overestimates the monoterpene concentration in winter. This can be clearly seen when plotting the ratio between $MT_{proxy1}$ and measured monoterpene concentration as a function of the day of the year (DOY) (Fig. 3). For other proxies this overestimation was not as clear. The overestimation of $MT_{proxy1}$ in winter time can be explained by the fact
that $MT_{proxy1}$ does not include the sink due to the oxidation of monoterpenes by $NO_3$. On the other hand, it can also be related to the seasonal variation of the emission potential of vegetation, described by the coefficient a in our proxy (see also Tarvainen et al., 2005; Aalto et al., 2015). To improve the seasonal variation of $MT_{proxy1}$, we fitted a DOY-dependent function to the ratio between $MT_{proxy1}$ and measurements (the red line in Fig. 3). Then, the corrected proxy was calculated from

$$[MT]_{proxy1.doy} = \frac{[MT]_{proxy1}}{\exp\left(h \times \cos\left(\frac{DOY}{365 \times 2\pi} + l\right) + m\right)} \tag{12}$$

Here parameters *h*, *l* and *m* have values of 0.38, 0.57 and 0.13.

**2.2.2 Proxy for monoterpene oxidation products**

The concentration of oxidation products of monoterpenes is governed by their production in the reactions between monoterpenes and different oxidants, and their removal by condensation on existing aerosol particles. The production rate can be calculated when knowing the concentrations of monoterpenes and different oxidants and the reaction rates between them.
The condensation sink (CS) can be calculated from the aerosol size distribution (Kulmala et al., 2001). Thus, by utilizing the proxies for monoterpene concentration derived in the previous section, the concentration of oxidation products of monoterpenes is obtained from

$$[OxOrg] = \frac{(k_{OH+MT}[OH] + k_{O3+MT}[O_3] + k_{NO3+MT}[NO_3]) \times MT_{proxy}}{CS}. \tag{13}$$



Here $k_{OH+MT}$, $k_{O3+MT}$ and $k_{NO3+MT}$ are reaction rate coefficients between monoterpenes and different oxidants, which were calculated as explained in the previous section, below Eq. (2). $MT_{proxy}$ is the concentration of monoterpenes based on the selected monoterpene proxy.

## 3 Results and discussion

### 3.1 Monoterpene proxy

The time series of the measured monoterpene concentration and different monoterpene proxies for the whole year 2013 and one week in September 2013 are illustrated in Fig. 4. All the proxies can be observed to follow the measured monoterpene concentration well in an annual scale, capturing the build-up of the concentration in spring, the maximum in summer, and the decrease in the concentration in autumn. In addition, the proxies seem to describe the daily variation of the concentration adequately. In this section, the ability of different monoterpene proxies to produce the seasonal and diurnal variation of monoterpene concentration is discussed in more detail.

### 3.1.1 Correlation between proxies and measurements

Figure 5 shows the correlation between different monoterpene proxies and the measured monoterpene concentration using data from the years 2006–2013. All the proxies correlated well with the measurements. One of the highest correlation coefficients (R = 0.74, $V_{90/10}$ = 6.6) was obtained for $MT_{proxy,ideal}$, in which the measured monoterpene concentration was used for calculating $NO_3$ concentration. This suggests that our equation for the proxy is plausible and considers the dynamics of the most important factors affecting the concentrations. On the other hand, from the true proxies, not using the monoterpene measurements, the best correlations were obtained for $MT_{proxy,simple}$ (R = 0.73, $V_{90/10}$ = 5.8) and $MT_{proxy1}$ (R = 0.70, $V_{90/10}$ = 7.0). Furthermore, for $MT_{proxy1,doy}$, a DOY-dependent version of $MT_{proxy1}$, the correlation coefficient was even higher, and the variation of the ratio between the measured and proxy concentrations, described by $V_{90/10}$ value, lower (R = 0.74, $V_{90/10}$ = 5.8). $MT_{proxy,simple}$ does not include any oxidation losses of monoterpenes, and $MT_{proxy1}$ and $MT_{proxy1,doy}$ include only the oxidation by OH and $O_3$. The fact that the highest correlation coefficients were still obtained for these proxies indicates that estimating the oxidation losses, without using the measured monoterpene concentration when calculating $NO_3$ concentration, introduces significant uncertainty into the proxy. From the proxies including the oxidation by $NO_3$, the correlation was stronger for $MT_{proxy2}$ (R = 0.68, $V_{90/10}$ = 6.9) than for $MT_{proxy3}$ (R = 0.65, $V_{90/10}$ = 9.2). In addition, for $MT_{proxy3}$ $V_{90/10}$-value was clearly higher than for any other proxy. This further suggests that when iteratively using the monoterpene proxy for calculating $NO_3$ concentration, as is done in $MT_{proxy3}$, the errors accumulate making the final proxy uncertain (overestimated monoterpene concentration leads to underestimation in $NO_3$ concentration, and thus in oxidation sink, which further increases the calculated monoterpene concentration).





It needs to be noted that there are more measured monoterpene concentration data available in spring and summer time than in other times of year (Table 1), and thus those data affect the correlation most when the whole data set is used. To investigate how well the proxies perform at different times of year, we studied the correlation between proxies and the measured monoterpene concentration in different seasons (Table 3). For all the proxies the correlation with the measurements was clear during spring, summer and autumn (R = 0.55–0.72), while in winter none of the proxies correlated with the measured monoterpenes concentration (R = -0.11–0.16). The variations of the ratio between the proxy and measurements, $V_{90/10}$ values, were also clearly higher for winter than for other seasons. Generally, the correlation coefficients were higher for $MT_{proxy,ideal}$, $MT_{proxy1}$, $MT_{proxy1,doy}$, $MT_{proxy,simple}$ than for $MT_{proxy2}$ and $MT_{proxy3}$ including all the oxidation processes. Interestingly, for the proxies not including the oxidation by $NO_3$, i.e. $MT_{proxy1}$, $MT_{proxy1,doy}$ and $MT_{proxy,simple}$, the highest correlation coefficients (even higher than for $MT_{proxy,ideal}$) were obtained in spring. However, in autumn the correlation coefficient was clearly highest for $MT_{proxy,ideal}$, which suggests that including the oxidation by $NO_3$ in the proxy is essential at that time of year. The weak correlation between measurements and all the proxies in wintertime can be due to several reasons. First of all, in winter biogenic emissions of monoterpenes are low (Hakola et al., 2012) and the concentrations are more affected by anthropogenic emissions which are not described by our proxies. At this time of year, measurement uncertainties are also high because the concentrations are often close to the detection limit of the PTR-MS (Taipale et al., 2008). On the other hand, in winter there are also more uncertainties related to proxies. For example, the boundary layer height is often not well defined in winter (Von Engeln and Teixeira, 2013). In addition, the contribution of $NO_3$ to the oxidation loss of monoterpenes can be expected to be higher in winter than in summer as there is less solar radiation (Peräkylä et al., 2014).

### 3.1.2 Monthly median concentrations

The monthly median concentrations of the measured monoterpenes and different proxies are shown in Fig. 6a. The measured monoterpene concentration was highest in July (median value $9.4\times10^9$ cm$^{-3}$) and lowest in February–March (median value $8.2\times10^8$ cm$^{-3}$). The summer maximum and the winter minimum were captured by all the proxies. However, the ratios between the concentrations predicted by the proxies and the measured monoterpene concentrations varied from month to month (Fig. 6b). In November–January, all the proxies overestimated the monoterpene concentration. $MT_{proxy,simple}$ was closest to the measurements, while $MT_{proxy1}$ overestimated the concentration most, showing the need for the DOY-dependent correction (see Sect. 2.2.1). In February, though, $MT_{proxy1}$ was close to the measurements together with $MT_{proxy1,doy}$ and $MT_{proxy,simple}$, while other proxies predicted too low concentrations. In March–May all the proxies performed pretty well, apart from $MT_{proxy,simple}$ overestimating the concentration in March. In midsummer, June–July, the proxy-to-measurement –ratios were close to one for all the proxies. In August all the proxies slightly overestimated the concentration. In September–October, the proxies were generally pretty close to the measurements; $MT_{proxy,simple}$ and $MT_{proxy1,doy}$ underestimated the monoterpene concentration and other proxies slightly overestimated them. Altogether, the median proxy-to-measurements ratios were between 0.8 and 1.3 in April–October and between 0.6 and 2.6 in November–March. The more detailed statistics of the ratio between the proxies and measured monoterpene concentration in different months are presented in Table A2 in Appendix.





All in all, it seems that the proxies generally predict too high concentrations in winter but are able to produce the correct concentration level relatively well in other seasons. In winter, $MT_{proxy,simple}$ tend to be closest to the measurements while at other times of the year $MT_{proxy,ideal}$, $MT_{proxy1,doy}$, $MT_{proxy2}$, and $MT_{proxy3}$ performed best. The overestimation of most of the proxies during wintertime may be related to the fact that, in reality, the emission potential of vegetation (described by the coefficient $a$ in our proxies) has a strong seasonal variation (Taipale et al., 2011; Rantala et al., 2015). For $MT_{proxy1}$ the DOY-dependent correction, which can be thought to represent the seasonal variation of the emission potential, improves the seasonal cycle of the proxy as $MT_{proxy1,doy}$ does not overestimate the winter-time concentrations as much as $MT_{proxy,1}$. The month-to-month variation of the proxy to measurement ratios may reflect the uneven distribution of measured data: for some months, there are measurements available only from few years, in which case the variation due to the specific conditions of those years strongly affects the proxy to measurements ratio (see Table 1).

### 3.1.3 Diurnal cycle

In addition to producing the correct concentration level at different times of years, it is essential that the proxies are able to describe the diurnal variation of monoterpene concentration. The median diurnal cycles of measured monoterpene concentrations and three proxies are illustrated in Fig. 7 for six different months (the rest of the months are shown in Fig. A1 in Appendix).

In March–September, the measured monoterpene concentration had a clear diurnal cycle with the lowest concentrations around noon and the highest concentrations at night or late in the evening. In March, $MT_{proxy1}$ and $MT_{proxy1,doy}$ captured the diurnal cycle of monoterpene concentration best. $MT_{proxy,simple}$ overestimated the concentration throughout the day, and $MT_{proxy,ideal}$, $MT_{proxy2}$ and $MT_{proxy3}$ predicted a too strong diurnal variation and too high daytime concentrations. In April and May all the proxies were able to produce the diurnal cycle quite well, having the daily maxima and minima around the same time as the measured concentration. In June–August, the measured monoterpene concentration had a very strong diurnal cycle with a minimum around noon. In these months, $MT_{proxy,simple}$ produced a clearly too weak diurnal cycle, while other proxies described the diurnal cycle of monoterpene concentration well. In September, the proxies performed adequately in general, except for $MT_{proxy,simple,}$ having a too weak diurnal cycle, and $MT_{proxy1,doy}$ predicting too low concentrations. In October–February, the measured monoterpene concentration had a significantly weaker diurnal cycle than in summer, and the highest concentrations were generally reached during daytime. In these months $MT_{proxy,ideal}$, $MT_{proxy2}$ and $MT_{proxy3}$ produced a too strong diurnal variation and clearly overestimated the concentration during daytime. $MT_{proxy1}$ also overestimated the concentration, while the concentrations predicted by $MT_{proxy,simple}$ and $MT_{proxy1,doy}$ were closest to the measurements.

Altogether, it seems that the proxies including all oxidation mechanisms (i.e. $MT_{proxy,ideal}$, $MT_{proxy2}$ and $MT_{proxy3}$) were able to describe the diurnal variation of monoterpene concentration well in summer when the diurnal cycle of the concentration was strong. The simpler proxies, especially $MT_{proxy,simple}$, were not able to capture the diurnal cycle as well at this time of year. On the other hand, in winter months, when the diurnal cycle was weaker, the simpler proxies produced the diurnal cycle best. The fact that the proxies were not able to produce the diurnal cycle accurately in winter is understandable, as at that time of year




the biogenic emissions of monoterpenes are low (see also the discussion in the end of the Sect. 3.1.1). Furthermore, in winter the relative role of $NO_3$ becomes higher (Peräkylä et al. 2014), as there is less solar radiation, and therefore the uncertainties related to calculating its concentration affect the proxies more than in summer. The boundary layer height, used in the proxies to describe the dilution of monoterpene concentration, is also not as well defined in winter as in summer (Von Engeln and
Teixeira, 2013).

### 3.2 Monoterpene oxidation products

The concentration of monoterpene oxidation products (OxOrg) in Hyytiälä was calculated for the years 1996–2014 (Fig. 8). $MT_{proxy1,doy}$ was used for the monoterpene concentration in the calculation, as it was observed to produce both the seasonal and diurnal cycle of monoterpene concentration reasonably well. In this section the seasonal and diurnal variations of the calculated
monoterpene oxidation products are discussed.

### 3.2.1 Seasonal variation

Figure 9 presents the monthly medians of the total concentration of monoterpene oxidation products and the contributions of different oxidants ($O_3$, OH and $NO_3$) to the total concentration during 1996–2014. The total concentration of oxidation products had a distinct seasonal cycle: the median concentrations were highest, $1.9–2.4\times10^8$ cm$^{-3}$, in summer (June–August) and lowest,
$3.4–5.4\times10^7$ cm$^{-3}$, in winter and early spring (January–March). Thus, the seasonal cycle of the oxidation products resembled the seasonal cycle of $MT_{proxy1,doy}$ (see Fig 6a). The summertime peak in the total concentration of oxidation products was caused by the oxidation products of $O_3$, which had a pronounced maximum in July and a minimum in February. The concentration of the oxidation products of $NO_3$, on the other hand, was lowest in spring (February–May) and highest in autumn and winter (October–January). In October–March the median concentrations of oxidation products of $NO_3$ were even higher
than the median concentrations of oxidation products of $O_3$. The oxidation products of OH had a clear seasonal cycle with the maximum in July and a minimum in winter, following the seasonal cycle of solar radiation. In summer months the median concentrations of oxidation products of OH were similar to the median concentrations of oxidation products of $NO_3$, both of them being clearly lower than the median concentrations of oxidation products of $O_3$. Thus, our proxy for the oxidation products of monoterpenes seems to be dominated by the oxidation of monoterpenes by $O_3$ in summer, while in winter the
oxidation by $NO_3$ is most significant.

### 3.2.2 Diurnal variation

In Fig. 10 the diurnal cycle of the concentration of monoterpene oxidation products (the total and the contributions of different oxidants) is illustrated for different seasons. In all seasons the total concentration of oxidation products was highest in the evening and lowest in the morning or around noon. The diurnal cycle was mostly determined by the diurnal variation in the
oxidation products of $NO_3$ in all seasons except summer (June–August). In March–May the total concentration of oxidation products stayed quite stable during daytime (from 6:00 to 18:00) and was at that time dominated by the oxidation products of





$O_3$. The concentration had a pronounced peak in the evening around 21:00, which was caused by the maximum in the concentration of the oxidation products of $NO_3$. In June–August the total concentration of oxidation products was lowest in the morning around 5:00, after which the concentration increased reaching its maximum around 21:00. The evening peak was mainly due to the maximum in the oxidation products of $O_3$, which dominated the total concentration throughout the day. On the other hand, at this time of year, the contribution of OH was also significant during daytime. In September–November the evening peak in the total concentration of oxidation products occurred earlier, around 18:00. It was primarily caused by the maximum in the concentration of the oxidation products of $NO_3$. During daytime the total concentration of oxidation products was dominated by the oxidation products of $O_3$. In December–February the total concentration of oxidation products followed the oxidation products of $NO_3$; the concentration was lowest during daytime and highest at night. In all seasons, except winter, the oxidation products of OH had a pronounced maximum around noon. At that time the concentration of the oxidation products of OH generally exceeded the concentration of the oxidation products of $NO_3$, being still lower than the concentration of the oxidation products of $O_3$. In winter, when there is only little solar radiation, the concentration of oxidation products of OH was very low throughout the day.

## 4 Conclusions

The oxidation products of monoterpenes likely have an important role in the formation and growth of aerosol particles in boreal forests (Kulmala et al., 1998; O'Dowd et al., 2002; Ehn et al., 2014; Jokinen et al., 2015). Therefore, the improved understanding of their concentration is needed, for example, when determining the climate effects of aerosol particles. In this study, we developed proxies for estimating the concentrations of monoterpenes and their oxidation products at a boreal forest site in Hyytiälä, southern Finland. For deriving and testing the validity of the proxies, we used monoterpene concentration measured in Hyytiälä during 2006–2013.

Our proxies for the monoterpene concentration include the temperature-driven emissions of monoterpenes, the dilution of the concentration caused by the mixing within the boundary layer, and the oxidation of monoterpenes by different oxidants (OH, $O_3$, and $NO_3$). Due to the difficulties related to estimating the concentration of $NO_3$, we tested five different versions of the proxy: 1) a proxy where the oxidation of monoterpenes by $NO_3$ is neglected, 2) a proxy where the oxidation of monoterpenes by $NO_3$ is neglected and an additional DOY-dependent correction is applied 3) a proxy where $NO_3$ concentration is estimated by using a constant value for monoterpene concentration, 4) a proxy where $NO_3$ concentration is calculated iteratively by using another monoterpene proxy, and 5) a proxy where all the oxidation processes are neglected.

All versions of the proxies for monoterpene concentration correlated well with the measured concentration (R = 0.65–0.74), and thus captured the seasonal variation of the monoterpene concentration. The best correlation with the measurements was obtained for the proxies not including the oxidation by $NO_3$, which suggests that estimating $NO_3$ concentration causes too much uncertainty to improve the performance of proxies. When investigating the ratios of the measured monoterpene concentration and the proxies, the proxies were mostly found to predict the correct concentration level in summer but



overestimate the concentration in winter. The typical proxy-to-measurements ratios were between 0.8 and 1.3 in summer and between 0.6 and 2.6 in winter. In addition, the proxies were observed to describe the diurnal variation of the monoterpene concentration reasonably well in summer but rather poorly in winter. Generally, the proxies including all the oxidation processes were able to produce the diurnal cycle of the monoterpene concentration in summer months when the measured

concentration had a strong diurnal variation. However, in winter, when the diurnal cycle of the measured concentration was weak, the simpler proxies were closer to the measurements. Altogether, the proxy neglecting the oxidation of monoterpenes by $NO_3$ and including a DOY-dependent correction ($MT_{proxy1,doy}$, see Eq. (12)) was found to describe the variation of monoterpene concentration most accurately. Therefore, we recommend using this proxy for predicting the monoterpene concentration in Hyytiälä and at similar, remote boreal forest sites.

To investigate the diurnal and seasonal variation of the oxidation products of monoterpenes in Hyytiälä, we calculated their concentration during 1996–2014 by using the most accurate monoterpene proxy. The oxidation products of monoterpenes had a clear seasonal cycle with the highest concentration in summer and the lowest concentration in winter. When studying the diurnal variation of the oxidation products, the concentration was found to be highest in the evening and lowest in the morning or around noon. The evening maximum was mainly caused by the oxidation products of $O_3$ in summer, and by the oxidation

products of $NO_3$ in other seasons. The contribution of the oxidation products of OH to the total concentration of oxidation products was highest in summer during daytime and minor in winter.

In the future, our proxies for the concentrations of monoterpenes and their oxidation products can be utilized, for example, when investigating the formation and growth of aerosol particles in Hyytiälä. The proxies could possibly be applied also at other measurement sites, at least those located in a boreal forest, but this remains to be tested in future studies. In addition,

further work is needed to validate the performance of the proxy for the monoterpene oxidation products by using the direct measurements of oxidized organic compounds.

**Acknowledgements**

This research was supported by the Academy of Finland (the Centre of Excellence program, grant nos. 1118615 and 272041, and the Academy professor project to M. Kulmala, grant no. 137749), the European Research Council Advanced Grant (ATM-

NUCLE, grant no. 227463), the EU FP7 research and innovation programme (ACTRIS-I3, grant no. 262254), and the CRAICC (Cryosphere–atmosphere interactions in a changing Arctic climate) project within the Nordic Top-level Research Initiative Programme 'Interaction between climate change and the cryosphere' (funded by Nordforsk). The work was also partly supported by the Office of Science (BER), U.S. Department of Energy via Biogenic Aerosols – Effects on Clouds and Climate (BAECC). The paper contributes to the Pan-Eurasian Experiment (PEEX) research agenda.



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





**Table 1. The number of data points (1-hour averages) of the measured monoterpene concentrations for each month during 2006–2013.**

| Month | Number of data points |
|-------|-----------------------|
| Jan | 368 |
| Feb | 588 |
| Mar | 801 |
| Apr | 812 |
| May | 857 |
| June | 1091 |
| July | 937 |
| Aug | 896 |
| Sep | 757 |
| Oct | 402 |
| Nov | 668 |
| Dec | 718 |

**Table 2. Parameters for different proxies.**

|   | $MT_{proxy.ideal}$ | $MT_{proxy1}$ | $MT_{proxy2}$ | $MT_{proxy3}$ | $MT_{proxy,simple}$ |
|---|---|---|---|---|---|
| a | $1.29 \times 10^7$ | $9.94 \times 10^6$ | $2.09 \times 10^7$ | $1.56 \times 10^7$ | $1.78 \times 10^{11}$ |
| b | 0.11 | 0.13 | 0.12 | 0.11 | 0.10 |
| c | -0.19 | -0.11 | -0.22 | -0.20 | -0.19 |
| d | -0.50 | -0.58 | -0.58 | -0.60 | -1.00 |





**Table 3.** Pearson correlation coefficient (R) and $V_{90/10}$-values between different proxies and the measured monoterpene concentration for all data and different seasons. The number of data points (N), of which the correlation coefficient was calculated, is shown in the last column.

|  | $MT_{proxy.ideal}$ | $MT_{proxy1}$ | $MT_{proxy1,doy}$ | $MT_{proxy2}$ | $MT_{proxy3}$ | $MT_{proxy.simple}$ | N |
|---|---|---|---|---|---|---|---|
| $R_{all}$ | 0.74 | 0.70 | 0.74 | 0.68 | 0.65 | 0.73 | 8191 |
| $V_{all}$ | 6.6 | 7.0 | 5.8 | 6.9 | 9.2 | 5.8 | |
| $R_{spring}$ | 0.66 | 0.71 | 0.72 | 0.64 | 0.55 | 0.69 | 2095 |
| $V_{spring}$ | 6.6 | 4.9 | 4.9 | 5.5 | 8.9 | 5.3 | |
| $R_{summer}$ | 0.69 | 0.69 | 0.71 | 0.60 | 0.61 | 0.60 | 2745 |
| $V_{summer}$ | 3.6 | 3.6 | 3.4 | 4.2 | 4.1 | 4.3 | |
| $R_{autumn}$ | 0.72 | 0.62 | 0.65 | 0.60 | 0.57 | 0.66 | 1766 |
| $V_{autumn}$ | 5.6 | 6.4 | 5.9 | 6.3 | 8.7 | 5.4 | |
| $R_{winter}$ | 0.16 | 0.01 | 0.01 | -0.11 | -0.08 | -0.03 | 1585 |
| $V_{winter}$ | 27.4 | 23.3 | 21.3 | 31.0 | 45.8 | 16.6 | |





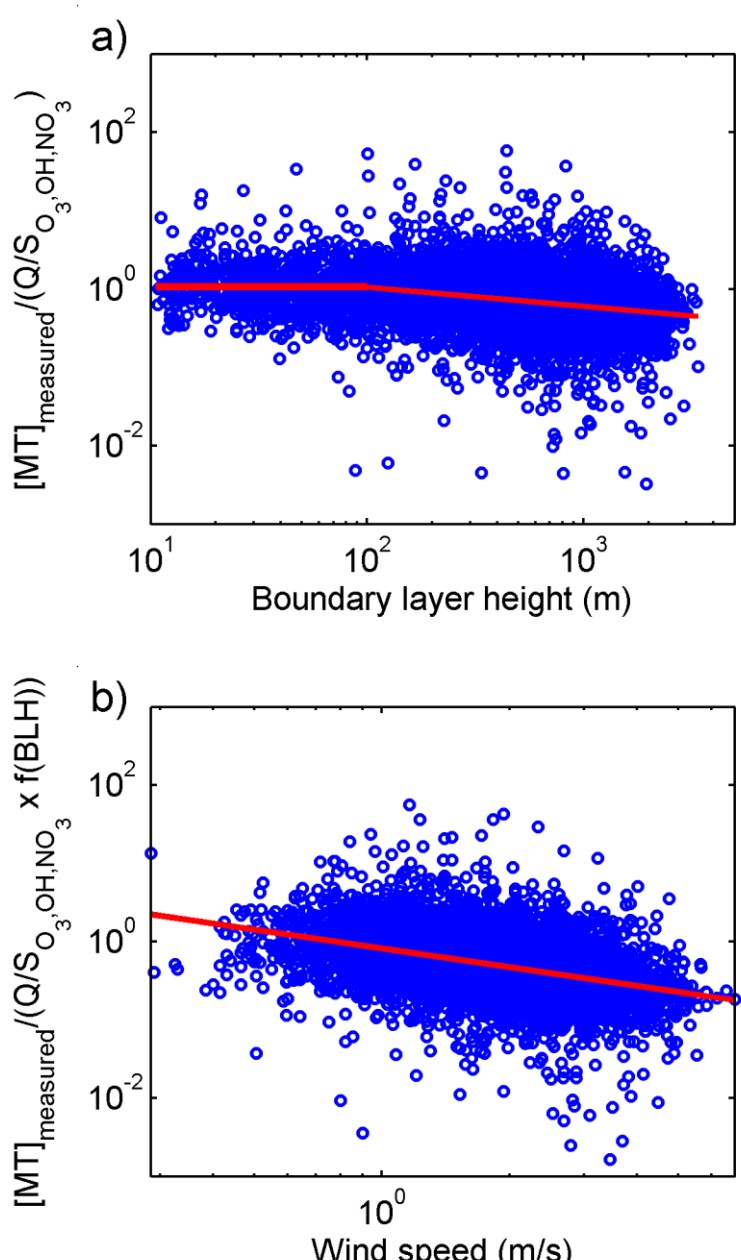

**Figure 1. The ratio between the measured monoterpene concentration and the initial version of MT$_{proxy,ideal}$ (a) as the function of boundary layer height, (b) as the function of wind speed after the dependence on the boundary layer height is already included. The red lines show the fitted functions.**





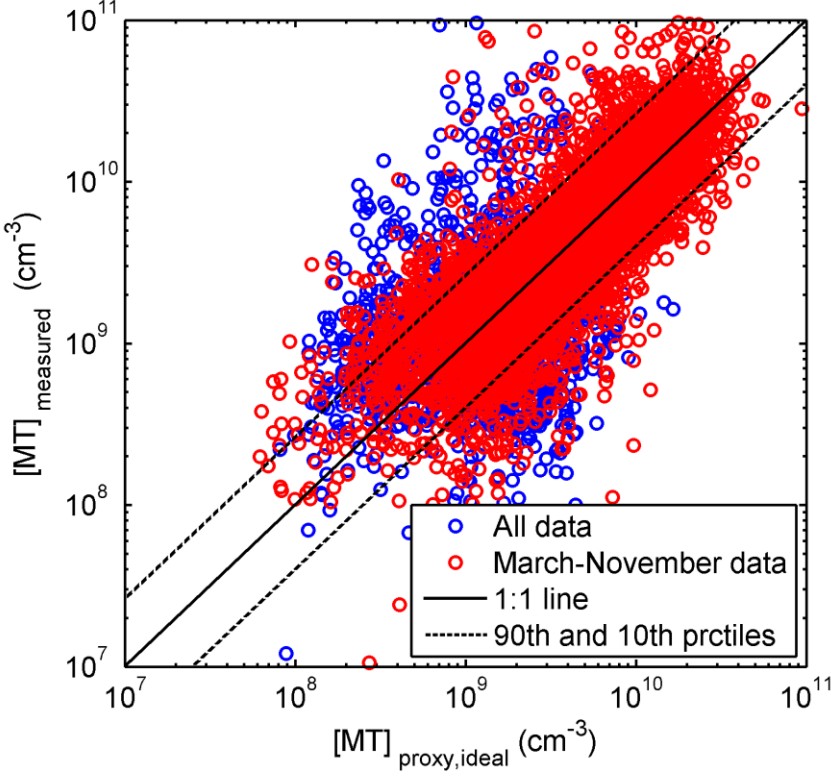

**Figure 2. A correlation between the measured monoterpene concentration and concentration predicted by MT$_{proxy,ideal}$, which is calculated by using the measured monoterpene concentration for determining the concentration of NO$_3$. The blue circles show all data and the red circles show data for March–November. The solid black line shows 1:1 line and the dotted black lines show 90th and 10th percentiles of the ratio between the measurements and proxy. The variability V$_{90/10}$ is calculated as the ratio of the 90th and 10th percentiles.**





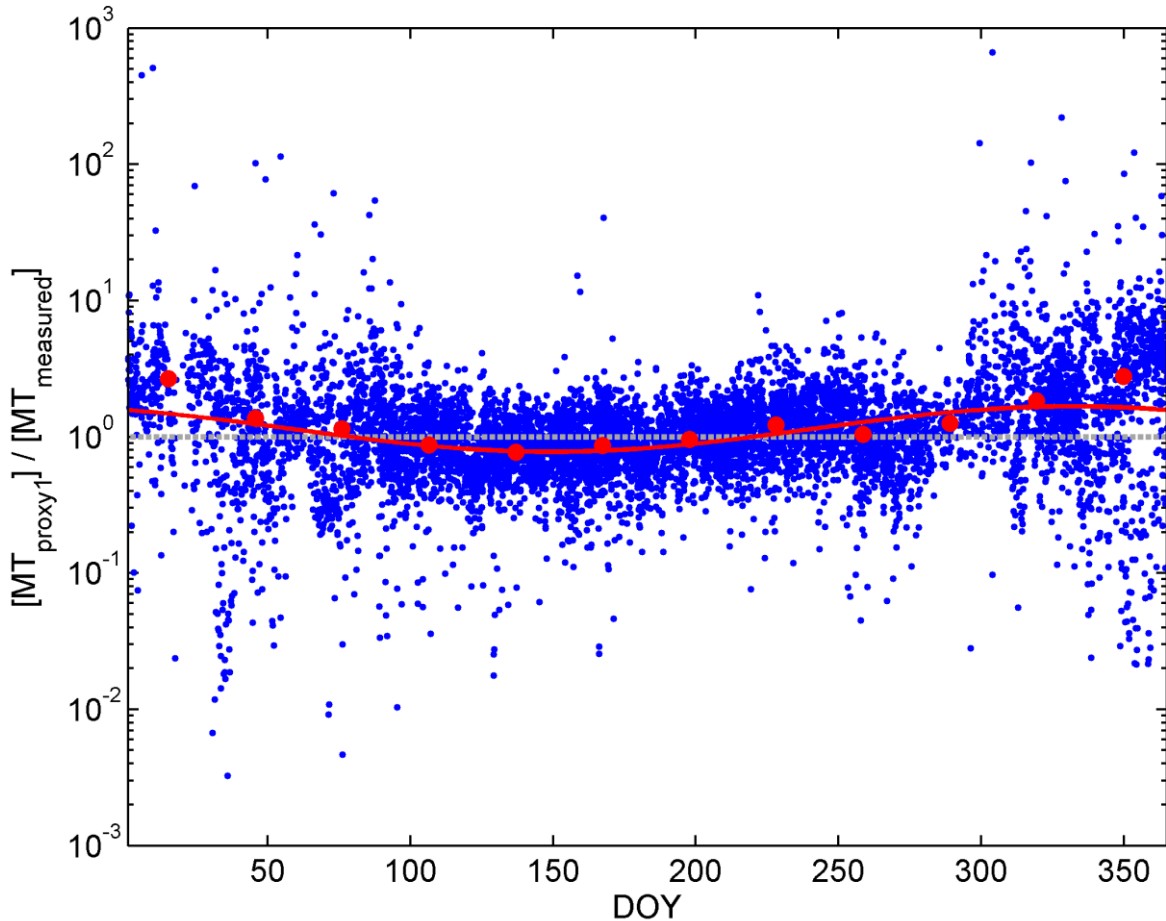

**Figure 3. The ratio between MT$_{proxy1}$ and the measured monoterpene concentration as a function of day of year (DOY). The red circles show the monthly medians of the ratio, and the red line depicts the function fitted to the ratio. The grey dashed line shows the ratio of one.**



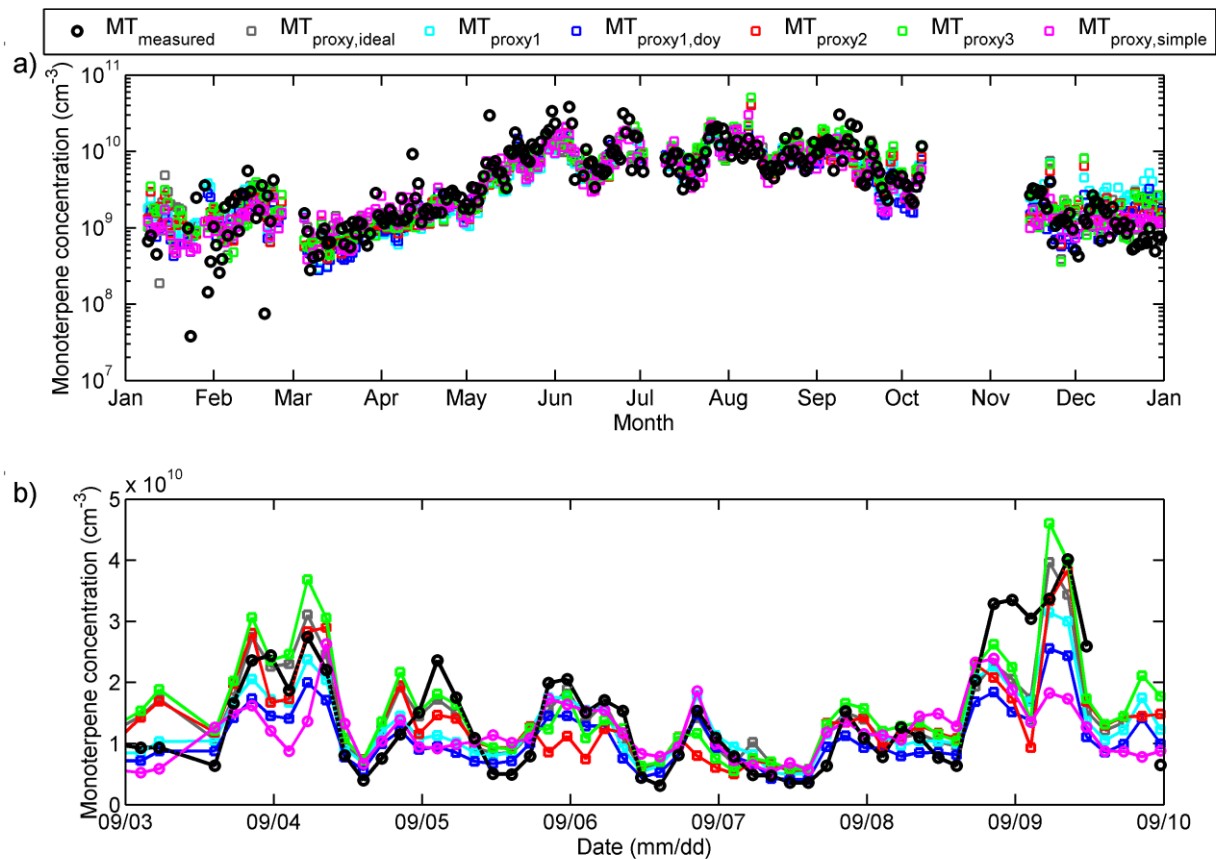

**Figure 4. Time series of the measured monoterpene concentration and the concentrations predicted by different proxies (a) for the whole year 2013, (b) for one week in September. The black circles show the measured concentration, the grey squares show MT$_{proxy,ideal}$, the light blue squares MT$_{proxy1}$, the dark blue squares MT$_{proxy1,doy}$, the red squares MT$_{proxy2}$, the green squares MT$_{proxy3}$,**
5  **and the magenta squares MT$_{proxy, simple}$.**



**Figure 5.** Correlations between the measured monoterpene concentration and the concentrations predicted by different proxies. The blue circles show all data and red circles data for March–November. The dotted line is 1:1 line. The correlation coefficients (R) and $V_{90/10}$-values are presented in the figures.

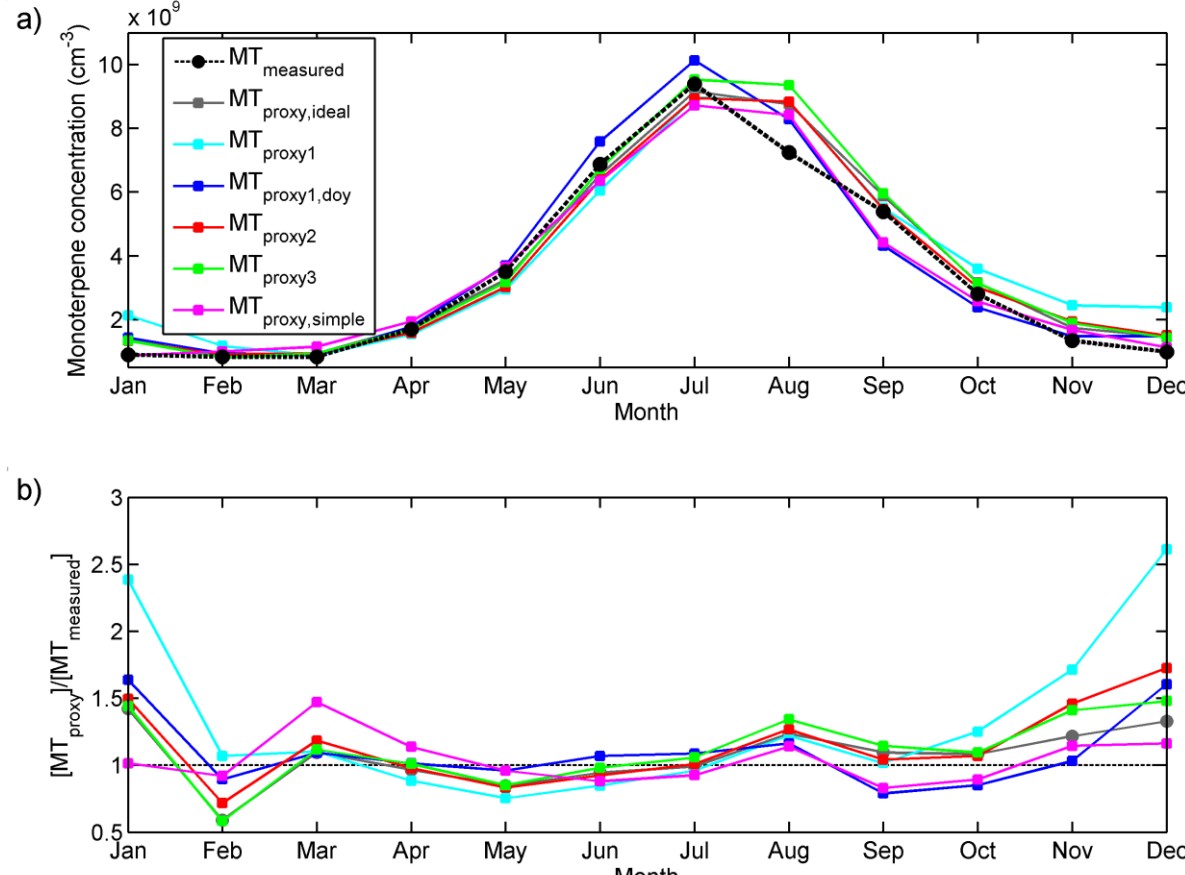

**Figure 6. Monthly medians of (a) the concentrations of the measured monoterpenes and different proxies and (b) the ratios of different proxies to measured monoterpene concentration. The black circles show the measured concentration, the grey squares show MT$_{proxy,ideal}$, the light blue squares MT$_{proxy1}$, the dark blue squares MT$_{proxy1,doy}$, the red squares MT$_{proxy2}$, the green squares MT$_{proxy3}$, and the magenta squares MT$_{proxy, simple}$.**





**Figure 7. Median diurnal variation of the measured monoterpenes and different proxies in different months (the rest of the months are shown in Fig. A1). The black circles show the measured concentration, the grey squares MT$_{proxy,ideal}$, the light blue squares MT$_{proxy1}$, the dark blue squares MT$_{proxy1,doy}$, the red squares MT$_{proxy2}$, the green squares MT$_{proxy3}$, and the magenta squares MT$_{proxy,simple}$. Note that the scales of the y-axis are not the same in all the figures.**



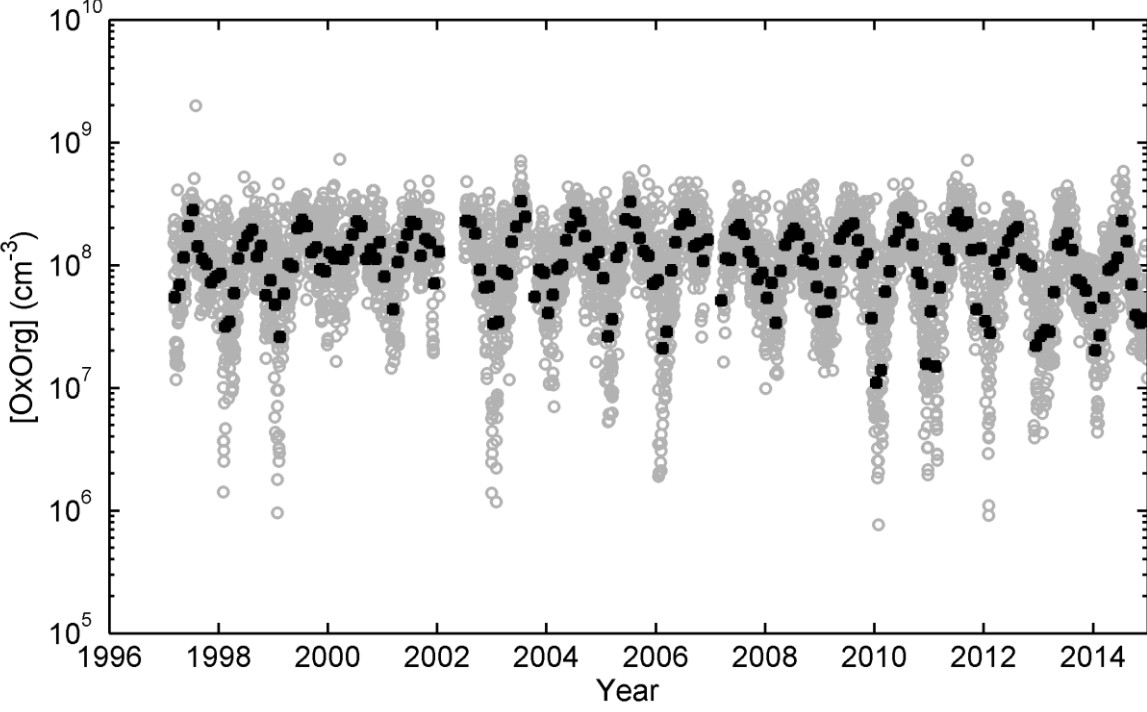

**Figure 8. The proxy for oxidation products of monoterpenes (OxOrg) during the years 1996–2014. The grey circles show the median concentration for each day and the black squares for each month.**





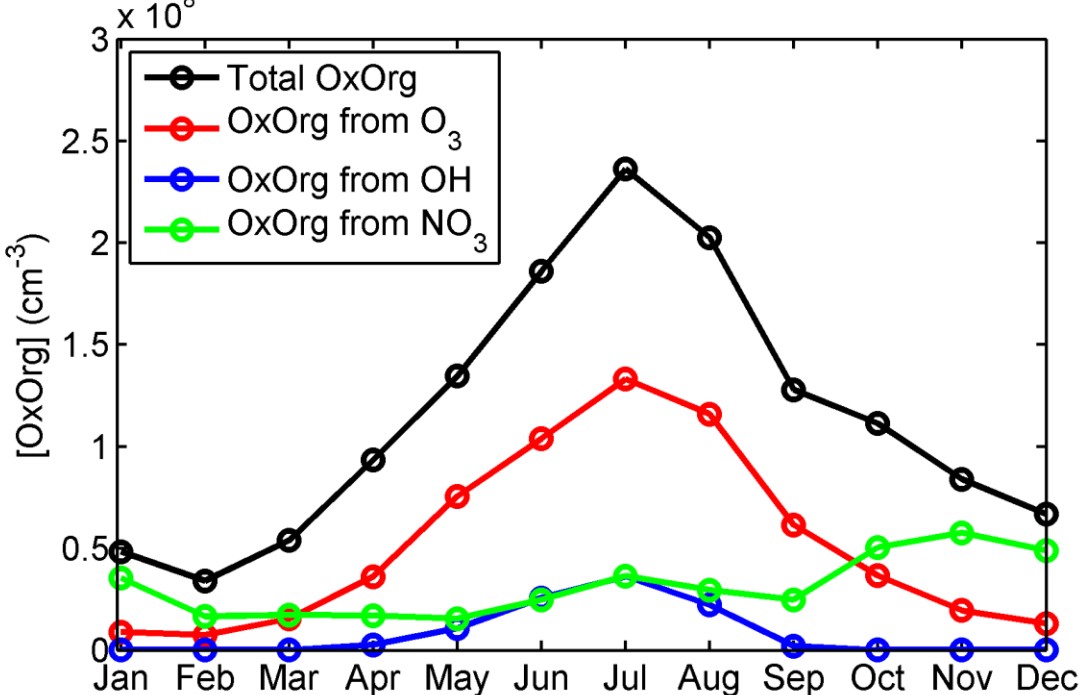

**Figure 9. Monthly medians of the proxy for the oxidation products of monoterpenes (OxOrg). The black line shows the total concentration of monoterpene oxidation products, the red line the oxidation products of monoterpenes from the oxidation by O₃, the blue line the oxidation products of monoterpenes from the oxidation by OH, and the green line the oxidation products of monoterpenes from the oxidation by NO₃.**



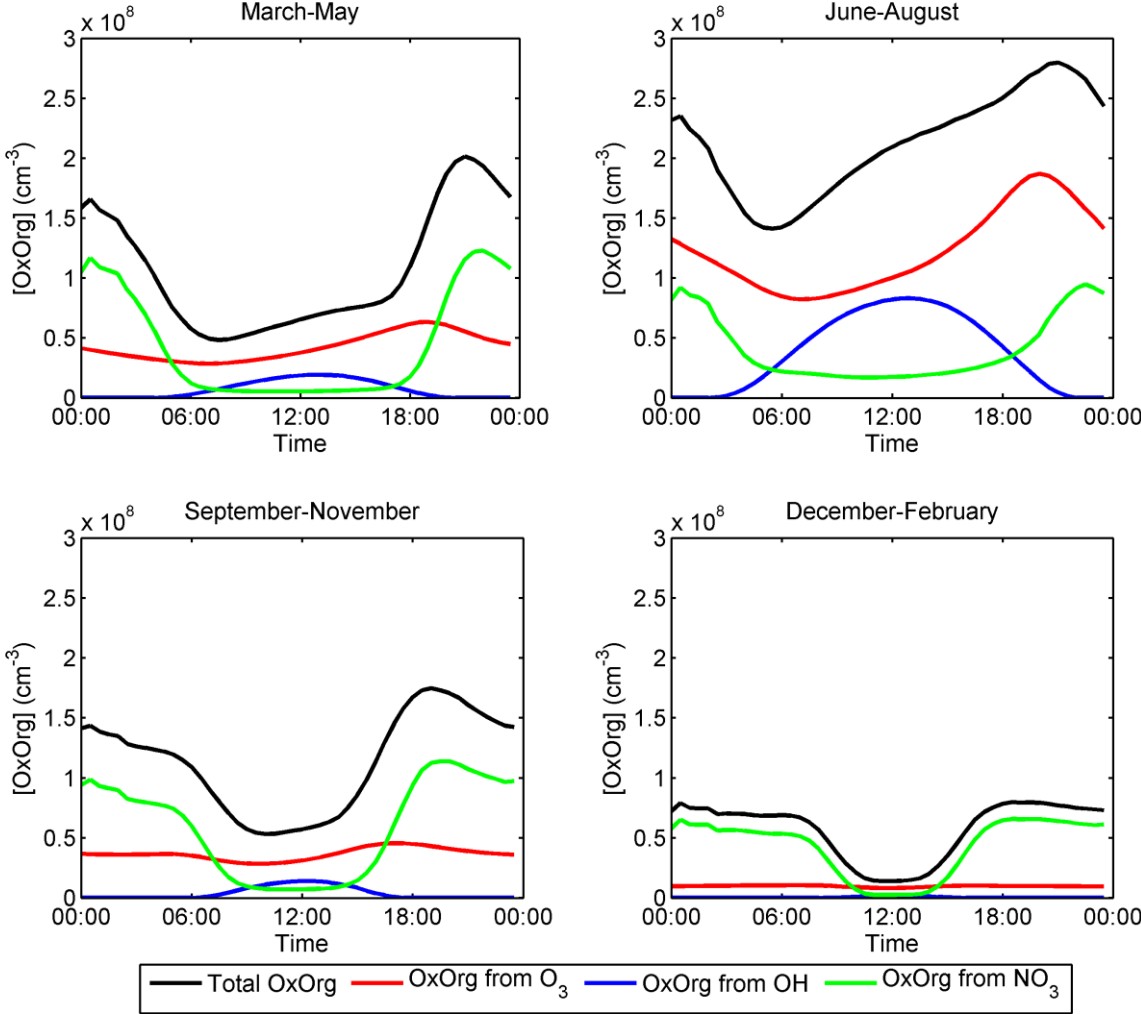

**Figure 10. Diurnal variation of the proxy for the oxidation products of monoterpenes (OxOrg) during different seasons. The black line shows the total concentration of oxidation products, the red line the oxidation products of monoterpenes from the oxidation by O₃, the blue line the oxidation products of monoterpenes from the oxidation by OH, and the green line the oxidation products of monoterpenes from the oxidation by NO₃.**

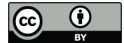



## Appendix

**Table A1. The temperature ($T$)-dependent relations of different reaction rate coefficients.**

| Rate coefficient | Temperature dependence | Reference |
|---|---|---|
| $k_{OH+MT}$ | $1.2 \times 10^{-11} \times \exp(440/T)$ | Atkinson et al. (2006) |
| $k_{O3+MT}$ | $6.3 \times 10^{-16} \times \exp(-580/T)$ | Atkinson et al. (2006) |
| $k_{NO3+MT}$ | $1.2 \times 10^{-12} \times \exp(490/T)$ | Atkinson et al. (2006) |
| $k_{O3+NO2}$ | $1.4 \times 10^{-13} \times \exp(-2470/T)$ | Atkinson et al. (2004) |
| $k_{NO3+NO}$ | $1.8 \times 10^{-11} \times \exp(110/T)$ | Atkinson et al. (2004) |




**Table A2.** The statistics of the ratio between the proxies and the measured monoterpene concentration in different months. The 10th, 25th, 50th, 75th, and 90th percentiles of the ratio are shown.

| Month | | $MT_{proxy.ideal}/$ $MT_{meas}$ | $MT_{proxy1}/$ $MT_{meas}$ | $MT_{proxy1.doy}/$ $MT_{meas}$ | $MT_{proxy2}/$ $MT_{meas}$ | $MT_{proxy3}/$ $MT_{meas}$ | $MT_{proxy.simple}/$ $MT_{meas}$ |
|---|---|---|---|---|---|---|---|
| Jan | 10th prctile | 0.32 | 0.66 | 0.46 | 0.35 | 0.24 | 0.33 |
| | 25th prctile | 0.67 | 1.44 | 0.95 | 0.77 | 0.62 | 0.68 |
| | 50th prctile | 1.42 | 2.39 | 1.64 | 1.49 | 1.44 | 1.01 |
| | 75th prctile | 2.89 | 3.79 | 2.57 | 2.62 | 2.94 | 1.46 |
| | 90th prctile | 5.24 | 5.30 | 3.61 | 4.44 | 5.66 | 2.26 |
| Feb | 10th prctile | 0.13 | 0.15 | 0.12 | 0.10 | 0.08 | 0.13 |
| | 25th prctile | 0.28 | 0.48 | 0.37 | 0.33 | 0.21 | 0.39 |
| | 50th prctile | 0.59 | 1.07 | 0.89 | 0.72 | 0.58 | 0.92 |
| | 75th prctile | 1.48 | 1.82 | 1.45 | 1.56 | 1.61 | 1.63 |
| | 90th prctile | 2.87 | 3.27 | 2.67 | 2.80 | 3.05 | 3.15 |
| Mar | 10th prctile | 0.24 | 0.30 | 0.29 | 0.28 | 0.18 | 0.41 |
| | 25th prctile | 0.46 | 0.59 | 0.60 | 0.60 | 0.43 | 0.85 |
| | 50th prctile | 1.10 | 1.10 | 1.09 | 1.18 | 1.12 | 1.47 |
| | 75th prctile | 2.16 | 1.85 | 1.85 | 2.17 | 2.44 | 2.44 |
| | 90th prctile | 3.71 | 3.26 | 3.34 | 3.67 | 3.92 | 4.20 |
| Apr | 10th prctile | 0.36 | 0.44 | 0.50 | 0.45 | 0.28 | 0.54 |
| | 25th prctile | 0.60 | 0.60 | 0.70 | 0.65 | 0.60 | 0.76 |
| | 50th prctile | 0.96 | 0.88 | 1.01 | 0.98 | 1.01 | 1.14 |
| | 75th prctile | 1.42 | 1.28 | 1.47 | 1.42 | 1.50 | 1.66 |
| | 90th prctile | 2.03 | 1.77 | 2.08 | 2.04 | 2.15 | 2.39 |
| May | 10th prctile | 0.45 | 0.42 | 0.54 | 0.45 | 0.39 | 0.50 |
| | 25th prctile | 0.64 | 0.56 | 0.71 | 0.63 | 0.65 | 0.69 |
| | 50th prctile | 0.85 | 0.75 | 0.96 | 0.83 | 0.85 | 0.96 |
| | 75th prctile | 1.16 | 1.03 | 1.32 | 1.15 | 1.21 | 1.31 |
| | 90th prctile | 1.53 | 1.36 | 1.73 | 1.48 | 1.59 | 1.80 |
| June | 10th prctile | 0.45 | 0.40 | 0.51 | 0.39 | 0.41 | 0.39 |
| | 25th prctile | 0.67 | 0.59 | 0.74 | 0.63 | 0.68 | 0.59 |
| | 50th prctile | 0.94 | 0.85 | 1.07 | 0.92 | 0.98 | 0.88 |
| | 75th prctile | 1.27 | 1.17 | 1.48 | 1.31 | 1.36 | 1.32 |
| | 90th prctile | 1.66 | 1.52 | 1.91 | 1.74 | 1.80 | 1.90 |





| | | | | | | | |
|------|--------------|------|------|------|------|------|------|
| July | 10th prctile | 0.52 | 0.53 | 0.61 | 0.51 | 0.53 | 0.51 |
| | 25th prctile | 0.73 | 0.74 | 0.83 | 0.74 | 0.75 | 0.67 |
| | 50th prctile | 0.99 | 0.96 | 1.09 | 1.01 | 1.06 | 0.92 |
| | 75th prctile | 1.31 | 1.28 | 1.43 | 1.36 | 1.42 | 1.24 |
| | 90th prctile | 1.73 | 1.64 | 1.84 | 1.79 | 1.88 | 1.70 |
| Aug | 10th prctile | 0.64 | 0.60 | 0.56 | 0.56 | 0.62 | 0.53 |
| | 25th prctile | 0.89 | 0.85 | 0.81 | 0.86 | 0.94 | 0.77 |
| | 50th prctile | 1.24 | 1.22 | 1.16 | 1.27 | 1.34 | 1.14 |
| | 75th prctile | 1.66 | 1.64 | 1.56 | 1.74 | 1.82 | 1.60 |
| | 90th prctile | 2.24 | 2.17 | 2.04 | 2.38 | 2.56 | 2.36 |
| Sep | 10th prctile | 0.50 | 0.48 | 0.36 | 0.41 | 0.41 | 0.40 |
| | 25th prctile | 0.73 | 0.71 | 0.55 | 0.65 | 0.71 | 0.57 |
| | 50th prctile | 1.09 | 1.01 | 0.79 | 1.04 | 1.14 | 0.83 |
| | 75th prctile | 1.60 | 1.52 | 1.21 | 1.57 | 1.78 | 1.20 |
| | 90th prctile | 2.23 | 2.13 | 1.73 | 2.25 | 2.54 | 1.76 |
| Oct | 10th prctile | 0.54 | 0.65 | 0.44 | 0.56 | 0.43 | 0.53 |
| | 25th prctile | 0.72 | 0.87 | 0.60 | 0.74 | 0.68 | 0.67 |
| | 50th prctile | 1.08 | 1.25 | 0.85 | 1.07 | 1.09 | 0.89 |
| | 75th prctile | 1.49 | 1.80 | 1.21 | 1.47 | 1.58 | 1.27 |
| | 90th prctile | 2.25 | 2.79 | 1.76 | 2.47 | 2.61 | 2.00 |
| Nov | 10th prctile | 0.41 | 0.57 | 0.34 | 0.43 | 0.31 | 0.43 |
| | 25th prctile | 0.61 | 1.05 | 0.63 | 0.83 | 0.59 | 0.72 |
| | 50th prctile | 1.22 | 1.71 | 1.03 | 1.46 | 1.41 | 1.14 |
| | 75th prctile | 2.15 | 3.04 | 1.85 | 2.40 | 2.47 | 1.97 |
| | 90th prctile | 3.69 | 5.28 | 3.17 | 4.24 | 4.63 | 3.42 |
| Dec | 10th prctile | 0.19 | 0.24 | 0.15 | 0.13 | 0.12 | 0.13 |
| | 25th prctile | 0.48 | 0.80 | 0.49 | 0.55 | 0.42 | 0.47 |
| | 50th prctile | 1.33 | 2.61 | 1.60 | 1.73 | 1.48 | 1.16 |
| | 75th prctile | 3.02 | 4.70 | 2.90 | 3.32 | 3.41 | 1.97 |
| | 90th prctile | 6.12 | 7.37 | 4.51 | 5.57 | 6.41 | 3.19 |



**Figure A1. Median diurnal variation of the measured monoterpenes and different proxies in different months (other months are shown in Fig. 7). The black circles show the measured concentration, the grey squares MT$_{proxy,ideal}$, the light blue squares MT$_{proxy1}$, the dark blue squares MT$_{proxy1,doy}$, the red squares MT$_{proxy2}$, the green squares MT$_{proxy3}$, and the magenta squares MT$_{proxy, simple}$.**