# Peer review of "Simple proxies for estimating the concentrations of monoterpenes and their oxidation products at a boreal forest site"

_Atmospheric Chemistry and Physics, 2016_

## Referee Comment (RC1) · Anonymous Referee #1 · 12 Jul 2016

The manuscript entitled "Simple proxies for estimating the concentrations of monoterpenes and their oxidation products at a boreal forest site" by Kontkanen et al., describes a method to calculate the mixing ratio of monoterpenes and their oxidation products using a simple proxy method. The method was applied to a huge data set from 2006 – 2013. I recommend publishing in ACPD after addressing the following issues.

General comments The manuscript is very interesting and well written. It contains a clear description of the proxy method. However, some information are still missing, in particular details about the PTR-MS measurements.

A clear description how the measured concentration of monoterpenes and their oxidation products was obtained is missing. The description should address the following

points: - What is the definition of [MT]measured? - Which compounds were considered for PTR-MS measurements to obtain [MT]measured? - How was the PTR-MS calibrated and which compounds were used for calibration? - Are the data of the PTR-MS corrected for temperature and RH dependency? This might be very important for summer and winter measurements. - How was the concentration of monoterpenes and oxidation products calculated? - Oxidation products have often a different response factor than their precursor compounds. Was this considered for calculation?

Minor comments

Page 3, line 10: The authors stated that few data were available which were obtained during measurements campaigns. Is there any comparison of the PTR-MS with other methods/instruments to validate the PTR-MS data?

Page 4, line 13: The approach considers reaction with O3, OH and NO3 as well as condensational sink. Is there a reason that the photolysis is not considered for the calculation? In particular, this might be very important for oxidation products such as pinonaldehyde, nopinone etc.

Page 28, Figure 9: According to the calculations O3 is the most important sink for monoterpenes as well as for monoterpene oxidation products. This is surprising as the rate constant for OH+monoterpene is in general much faster than with O3. What is the reason for this observation? Furthermore, it was stated that a-pinene is the most important monoterpene. The first-generation oxidation product is pinonaldehyde. As pinonaldehyde does not contain any C-C double bond is cannot react with O3. This is the same for few other oxidation products like nopinone. Why is OH radical reaction not considered for monoterpene oxidation products?

―――――――――――――――――

---

## Referee Comment (RC2) · Anonymous Referee #2 · 11 Aug 2016

The authors present a long-term dataset of proton transfer reaction mass spectrometer (PTR-MS) measurements of monoterpene concentrations at the SMEAR II station in Hyytiala, Finland. The authors then derive a series of proxies by which monoterpene concentrations, and those of organic oxidation products, may be estimated – enabling the generation of longer datasets at this site and potentially estimates of these values at sites where these measurements are not available.

The study highlights the importance of monoterpene oxidation by different oxidants at different times of the year and these datasets will be valuable in terms of evaluating the performance of regional and global models that aim to represent processes involving monoterpenes and their oxidation products.

[Figure]

The paper is very well written, clearly structured and very thorough. The proxies used are clearly derived and their performance is well evaluated. I would recommend this manuscript for publication in ACP and only have the following very minor comments and technical suggestions.

Specific / Minor Comments:

The authors demonstrate that estimating the NO3 concentration introduces uncertainty to the monoterpene concentration proxies, that the correlation (with measured monoterpene concentrations) is poorest during the winter when oxidation by NO3 would dominate, and that the majority of the oxidation products are being generated via NO3 oxidation outside of winter. Perhaps the authors could add a comment to the conclusions about the need for accurate NO3 measurements (or measurement systems with a lower detection limit)?

In the Conclusions the authors mention the application of the proxies at other boreal sites, this is a study I will be very interested to read as I think their success in predicting the monoterpene concentrations at other sites will be the true test of the proxies. Specifically, it will be useful to examine the impact of applying the DOY-dependent function at other sites.

Technical Suggestions:

p1, line 28: replace volatile with "volatility"

p9, line 27 and 30: perhaps replace "pretty well" and "pretty close" with something more specific/scientific?

———————————————————

---

## Author Comment (AC1) · 6 Oct 2016

**Reply to Referee #1**

We thank Anonymous Referee #1 for their helpful comments. We have answered to the comments below. The bold text is quoted from the referee's comments, and the text in italics has been added to the manuscript.

**GENERAL COMMENTS**

**The manuscript entitled "Simple proxies for estimating the concentrations of monoterpenes and their oxidation products at a boreal forest site" by Kontkanen et al., describes a method to calculate the mixing ratio of monoterpenes and their oxidation products using a simple proxy method. The method was applied to a huge data set from 2006–2013. I recommend publishing in ACPD after addressing the following issues.**

**The manuscript is very interesting and well written. It contains a clear description of the proxy method. However, some information are still missing, in particular details about the PTR-MS measurements. A clear description how the measured concentration of monoterpenes and their oxidation products was obtained is missing. The description should address the following points:**

**1) What is the definition of [MT]$_{measured}$?**

**2) Which compounds were considered for PTR-MS measurements to obtain [MT]$_{measured}$?**

**3) How was the PTR-MS calibrated and which compounds were used for calibration?**

**4) Are the data of the PTRMS corrected for temperature and RH dependency? This might be very important for summer and winter measurements.**

**5) How was the concentration of monoterpenes and oxidation products calculated?**

**6) Oxidation products have often a different response factor than their precursor compounds. Was this considered for calculation?**

As suggested by the referee we added a more detailed description of the PTR-MS measurements in Section 2.1:

*The PTR-MS was maintained at a drift tube pressure of 1.95–2.20 mbar. The primary ion signal ($H_3O^+$) varied between 1 and $30 \times 10^6$ cps, being typically around $10 \times 10^6$ cps. With these settings, the E/N ratio where E is the electric field and N the number density of the gas in the drift tube, varied between 105 and 125 Td (Td = $10^{-21}$ V m$^{-2}$). The instrumental background was determined every second or third hour with a zero-air generator (Parker ChromGas Zero Air Generator, model 3501, USA), and the instrument was calibrated every 2–4 weeks using an alpha-pinene standard gas (Apel-Riemer Environmental Inc., USA, or Ionimed GmbH, Austria) which was diluted to around 1–5 ppbv. The monoterpene concentrations were derived from the measured m/z (mass-to-charge ratio) 137 signal according to Taipale et al. (2008). Shortly, the measured signal was first normalized using measured $H_3O^+$ and $H_2OH_3O^+$ signals, and the drift tube temperature and pressure. Then, the normalized signal was converted to the volume mixing ratio using a normalized instrumental sensitivity.*

In addition, the answers to the referee's specific questions are below:

1) [MT]$_{measured}$ is defined as the measured monoterpene concentration (see the next point) in units molecules cm$^{-3}$.

2) The measured monoterpenes are detected at *m/z* 137. This sums up all monoterpenes as well as other compounds with molecular mass of 136 amu and proton affinity higher than that of water because the PTR-MS is unable to distinguish between compounds with the same molecular mass.

3) The calibration gas included alpha-pinene as a representative monoterpene. It is well justified since alpha-pinene is the most common monoterpene found in ambient air in boreal forest (e.g. Hakola et al., 2009). The calibration was conducted as described in Taipale et al. (2008). In short, calibration gas containing 1 ppmv of alpha-pinene was diluted close to typical atmospheric concentrations using zero air generator based on catalytic conversion (Parker ChromGas Zero Air Generator, model 3501, USA). The calibration was conducted 2–4 times per month.

4) PTR-MS data were not corrected for temperature and RH. Ambient air was drawn to the PTR-MS through a heated sampling line, and the measured monoterpene concentration was considered to represent the ambient monoterpene concentration at the measurement height under all temperature and RH conditions.

5) The concentration of monoterpenes was calculated following the scheme described in Taipale et al. (2008). The measurements of monoterpenes oxidation products were not utilized in this study.

6) As mentioned in the previous point, the measurements of monoterpenes oxidation products were not used.

**MINOR COMMENTS**

**Page 3, line 10: The authors stated that few data were available which were obtained during measurements campaigns. Is there any comparison of the PTR-MS with other methods/instruments to validate the PTR-MS data?**

In Ruuskanen et al. (2005) PTR-MS measurements were compared with GC-MS (gas-chromatograph mass spectrometer) measurements at the SMEAR II station. A reasonable agreement was found between these methods in monoterpene concentrations over a period of several months. More recently, Kajos et al. (2015) compared the concentrations of oxidized and aromatic VOCs (methanol, acetaldehyde, acetone, benzene and toluene) measured by two PTR-MS and two GC-MS instruments in ambient air at the same site. A very good correlation between different methods was obtained for benzene and acetone.

**Page 4, line 13: The approach considers reaction with $O_3$, OH and $NO_3$ as well as condensational sink. Is there a reason that the photolysis is not considered for the calculation? In particular, this might be very important for oxidation products such as pinonaldehyde, nopinone etc.**

Regarding this comment, as well as the comments below, it is important to note that in our calculation we do not consider any reactions for the first-generation oxidation products but we assume that they are further oxidized until they are condensable. Therefore, we also did not include the photolysis of oxidation products in the calculation. This was, however, not clearly enough explained in the text and therefore we added the following sentences to the manuscript in Section 2.2.2:

*It should be noted that OxOrg can be thought to represent the total concentration of oxidized monoterpenes, because it takes into account all the generations of oxidation products, from the first oxidation until condensable molecules. However, as the formulation of this proxy presumes that oxidation takes place relatively fast and that there are no others sinks than condensation sink, it should be considered as a rough estimate for the concentration of condensable organic vapors.*

**Page 28, Figure 9: According to the calculations $O_3$ is the most important sink for monoterpenes as well as for monoterpene oxidation products. This is surprising as the rate constant for**

**OH+monoterpene is in general much faster than with O₃. What is the reason for this observation?**

It is true that these results indicate that the oxidation by $O_3$ is the most important sink for monoterpenes. However, this can be explained by the fact that $O_3$ concentration is orders of magnitude higher than OH concentration. The median OH concentration (estimated from a proxy for times when there was solar radiation) was $2.4 \times 10^5$ cm$^{-3}$ while the median $O_3$ concentration for the same times was $8.2 \times 10^{11}$ cm$^{-3}$. Regarding monoterpene oxidation products, we do not analyze the relative importance of their reactions, as explained above.

**Furthermore, it was stated that a-pinene is the most important monoterpene. The first-generation oxidation product is pinonaldehyde. As pinonaldehyde does not contain any C-C double bond it cannot react with O₃. This is the same for few other oxidation products like nopinone. Why is OH radical reaction not considered for monoterpene oxidation products?**

As explained above, we do not consider any reactions for the first-generation oxidation products.

**REFERENCES**

Hakola, H., Hellén, H., Tarvainen, V., Bäck, J., Patokoski, J., and Rinne, J.: Annual variations of atmospheric VOC concentrations in a boreal forest, Boreal Environ. Res., 14, 722–730, 2009.

Kajos, M. K., Rantala, P., Hill, M., Hellén, H., Aalto, J., Patokoski, J., Taipale, R., Hoerger, C. C., Reimann, S., Ruuskanen, T. M., Rinne, J., and Petäjä, T.: Ambient measurements of aromatic and oxidized VOCs by PTR-MS and GC-MS: intercomparison between four instruments in a boreal forest in Finland, Atmos. Meas. Tech., 8, 4453-4473, doi:10.5194/amt-8-4453-2015, 2015.

Ruuskanen, T. M., Kolari, P., Bäck, J., Kulmala, M., Rinne, J., Hakola, H., Taipale, R., Raivonen, M., Altimir, N., and Hari, P.: On-line field measurements of monoterpene emissions from Scots pine by proton-transfer-reaction mass spectrometry, Boreal Environ. Res., 10, 553–567, 2005

Taipale, R., Ruuskanen, T. M., Rinne, J., Kajos, M. K., Hakola, H., Pohja, T., and Kulmala, M.: Technical Note: Quantitative long-term measurements of VOC concentrations by PTR-MS – measurement, calibration, and volume mixing ratio calculation methods, Atmos. Chem. Phys., 8, 6681-6698, doi:10.5194/acp-8-6681-2008, 2008.

---

## Author Comment (AC2) · 6 Oct 2016

**Reply to Referee #2**

We thank Anonymous Referee #2 for their helpful comments. We have answered to the comments below. The bold text is quoted from the referee's comments, and the text in italics has been added to the manuscript.

**GENERAL COMMENTS**

**The authors present a long-term dataset of proton transfer reaction mass spectrometer (PTR-MS) measurements of monoterpene concentrations at the SMEAR II station in Hyytiala, Finland. The authors then derive a series of proxies by which monoterpene concentrations, and those of organic oxidation products, may be estimated – enabling the generation of longer datasets at this site and potentially estimates of these values at sites where these measurements are not available. The study highlights the importance of monoterpene oxidation by different oxidants at different times of the year and these datasets will be valuable in terms of evaluating the performance of regional and global models that aim to represent processes involving monoterpenes and their oxidation products.**

**The paper is very well written, clearly structured and very thorough. The proxies used are clearly derived and their performance is well evaluated. I would recommend this manuscript for publication in ACP and only have the following very minor comments and technical suggestions.**

**MINOR COMMENTS**

**The authors demonstrate that estimating the $NO_3$ concentration introduces uncertainty to the monoterpene concentration proxies, that the correlation (with measured monoterpene concentrations) is poorest during the winter when oxidation by $NO_3$ would dominate, and that the majority of the oxidation products are being generated via $NO_3$ oxidation outside of winter. Perhaps the authors could add a comment to the conclusions about the need for accurate $NO_3$ measurements (or measurement systems with a lower detection limit)?**

As suggested by the referee, we added the following comment in the conclusions (page 12, line 31 in the ACPD version):

*"...thus demonstrating the need for direct measurements of $NO_3$ concentration with the low enough detection limit."*

**In the Conclusions the authors mention the application of the proxies at other boreal sites, this is a study I will be very interested to read as I think their success in predicting the monoterpene concentrations at other sites will be the true test of the proxies. Specifically, it will be useful to examine the impact of applying the DOY-dependent function at other sites.**

**TECHNICAL SUGGESTIONS**

**p1, line 28: replace volatile with "volatility"**

We made this change.

**p9, line 27 and 30: perhaps replace "pretty well" and "pretty close" with something more specific/scientific?**

We changed the "pretty well" on line 27 to "*adequately*" and "pretty close" on line 30 to "*reasonably close*".